# Detection of Unknown and Rare Pathogenic Variants in Antithrombin, Protein C and Protein S Deficiency Using High-Throughput Targeted Sequencing

**DOI:** 10.3390/diagnostics12051060

**Published:** 2022-04-23

**Authors:** Petr Vrtel, Ludek Slavik, Radek Vodicka, Julia Stellmachova, Martin Prochazka, Jana Prochazkova, Jana Ulehlova, Peter Rohon, Tomas Simurda, Jan Stasko, Ivana Martinkova, Radek Vrtel

**Affiliations:** 1Department of Medical Genetics, University Hospital Olomouc, 77900 Olomouc, Czech Republic; petr.vrtel@fnol.cz (P.V.); julia.stellmachova@fnol.cz (J.S.); martin.prochazka@fnol.cz (M.P.); peter.rohon@upol.cz (P.R.); 2Department of Hemato-Oncology, University Hospital Olomouc, 77900 Olomouc, Czech Republic; jana.prochazkova@fnol.cz (J.P.); jana.ulehlova@fnol.cz (J.U.); 3National Centre of Hemostasis and Thrombosis, Department of Hematology and Transfusiology, Comenius University in Bratislava, Jessenius Faculty of Medicine in Martin and University Hospital in Martin, 03659 Martin, Slovakia; tsimurda@orava.sk (T.S.); stasko@jfmed.uniba.sk (J.S.); 4HEMACENTRUM spol s.r.o., 32600 Pilsen, Czech Republic; martinkova.ivana@hemacentrum.cz

**Keywords:** high-throughput sequencing, NGS, antithrombin deficiency, protein C deficiency, protein S deficiency, mutation detection rate, anticoagulant

## Abstract

The deficiency of natural anticoagulants—antithrombin (AT), protein C (PC), and protein S (PS)—is a highly predisposing factor for thrombosis, which is still underdiagnosed at the genetic level. We aimed to establish and evaluate an optimal diagnostic approach based on a high-throughput sequencing platform suitable for testing a small number of genes. A fast, flexible, and efficient method involving automated amplicon library preparation and target sequencing on the Ion Torrent platform was optimized. The cohort consisted of a group of 31 unrelated patients selected for sequencing due to repeatedly low levels of one of the anticoagulant proteins (11 AT-deficient, 13 PC-deficient, and 7 PS-deficient patients). The overall mutation detection rate was 67.7%, highest in PC deficiency (76.9%), and six variants were newly detected—*SERPINC1* c.398A > T (p.Gln133Leu), *PROC* c.450C > A (p.Tyr150Ter), c.715G > C (p.Gly239Arg) and c.866C > G (p.Pro289Arg), and *PROS1* c.1468delA (p.Ile490fs) and c.1931T > A (p.Ile644Asn). Our data are consistent with those of previous studies, which mostly used time-consuming Sanger sequencing for genotyping, and the indication criteria for molecular genetic testing were adapted to this process in the past. Our promising results allow for a wider application of the described methodology in clinical practice, which will enable a suitable expansion of the group of indicated patients to include individuals with severe clinical findings of thrombosis at a young age. Moreover, this approach is flexible and applicable to other oligogenic panels.

## 1. Introduction

Thromboembolic disease is characterized by the occurrence of blood clots (thrombi) in blood vessels. Antithrombin (AT), protein C (PC), and protein S (PS) deficiency are serious factors in venous thromboembolism (VTE). VTE is characterized by the formation of a thrombus in the venous system, which can subsequently be complicated by embolization into the pulmonary artery (PE). Thrombosis is a common pathological condition that plays an important role in public health.

The annual incidence of VTE in a population of European origin ranges from 104 to 183 cases per 100,000 people [1]. One-year mortality rates of 16.7%–25% have been described in cohort studies of patients. The rate of cancer-related thromboembolism increases the risk of death [2,3,4]. The occurrence of thromboembolic events has a multifactorial basis. The genetic component of susceptibility to thrombosis has been established at more than 60% (in selected patients with first manifestation of thrombosis under 45 years of age) [5]. Acquired factors such as surgery, use of oral contraceptives, immobilization, pregnancy, smoking, age, and cancer can trigger an episode of thrombosis [5]. Recently, a possible association of a higher incidence of VTE in patients hospitalized with COVID-19 infection was described [6].

### 1.1. Factor V Leiden and Prothrombin Mutation

Mutations in genes that code for proteins involved in the blood clotting system are responsible for hereditary thrombophilia. The single nucleotide polymorphism (SNP) of factor V Leiden and prothrombin factor II G20210A is well-known and routinely tested due to its high population frequency in the general European population. FV Leiden with a population frequency of 2–15% (2.7% according to gnomAD-Genomes) has been found in 15–25% of patients with deep vein thrombosis (DVT) [7,8,9]. The prothrombin G20210A mutation has been found in 1–4% of healthy European individuals and in 6–16% of patients with unselected DVT. Heterozygous Leiden mutation carriers with a lifelong hypercoagulable state have a 2–8 times higher risk of VTE. This risk is 50–80 times higher in homozygotes. The prothrombin G20210A mutation can cause hypercoagulability and a 2–4-fold increased risk of VTE [8,9]. These variants show autosomal dominant genetic status with incomplete penetrance.

### 1.2. Anticoagulant Proteins

Less common but severe genetic risk factors for thrombosis are deficiencies of the anticoagulant proteins AT, PS, and PC, and genetic causes are mutations in the genes that encode these key anticoagulant proteins—*SERPINC1*, *PROS1*, and *PROC*, respectively. The occurrence of these anticoagulant deficiencies is not as common as the Leiden and prothrombin variants, but they have the same or higher risk of thrombosis for their carriers. The prevalence of AT, PC, and PS deficiencies in the European population has been estimated to be from 0.02% to 0.2%, from 0.2% to 0.3%, and 0.5%, respectively [8,10,11]. In cohorts of patients with thrombosis, these deficits have been found to be from 1% to 2%, up to 5%, and 2.2%, respectively [8,10,11,12]. In selected patients with VTE and aged <45 years, the prevalence is significantly higher at up to 4.9%, 8.6%, and 7.5%, respectively [12]. Di Minno et al. 2015 [13] provided a meta-analytical study in which they established a 16-fold increased risk of VTE in AT-deficient patients, whereas Dahlbäck 2008 [8] calculated this risk as less than 10-fold. A 5–10-fold increased risk of VTE has been established in PC deficiency [13,14]. A similar situation has been described for PS deficiency, with a 5–10 times higher risk of VTE compared to the general population [11,13,14]. Mutations in the genes encoding these anticoagulant proteins mostly show autosomal dominant inheritance, often with reduced penetrance and variable expression. Rarely, an autosomal recessive pattern has also been described. See OMIM database under numbers: 613118; 176880; 612283 (www.omim.org accessed on 26 February 2022).

### 1.3. Genetic Testing for Genetic Suspicion of Thrombophilia

People who meet at least one of the following criteria are indicated for genetic testing of hereditary thrombophilias: onset of idiopathic VTE at age < 50 years, VTE in an unusual location, young patients with arterial ischemia due to paradoxical embolism, and first episode of VTE with a positive family history of VTE. In women, further criteria include: VTE during puerperium or pregnancy, VTE during use of oral contraceptives or hormone replacement, VTE before use of hormonal replacement, women with multiple unexplained pregnancy losses, and young women with a positive family history of VTE before prescription of oral contraceptives [15].

The method of genetic testing depends on the presumed type of molecular genetic defect. In the case of FV Leiden and G20210A polymorphisms, SNP detection methods such as Taq-Man Real-Time PCR, allele-specific PCR, SNaPshot, and Strip Assay are used. If the genetic involvement of AT, PS, and PC deficiency is suspected, a different approach is required. Because causative genes lack mutational hotspots, the primary goal of testing is to look for single nucleotide variants (SNVs) that may be randomly localized throughout the gene. The first and still frequently used method is Sanger sequencing complemented by MLPA to detect possible copy number variation (CNV) [10,11]. The best approach to detect causal gene variants in anticoagulation deficient probands is probably the high-throughput sequencing (HTS) methodology. This approach helps to expand testing capacity and streamline the search for rare genetic variants. In 2012, the results of one of the next-generation sequencing (NGS) pilot panels for the detection of genetic variants were published [16].

The aims of this study were to establish and evaluate an optimal diagnostic procedure based on the HTS platform that is suitable for testing a small number of genes.

In this paper, we present the methodology and the first results of a high-throughput gene panel for the detection of rare genetic variants in thrombophilic conditions. We perform this gene panel for the routine diagnosis of patients with suspected hereditary AT, PC, or PS deficiency. This was probably the first application of NGS to this issue in the Czech Republic.

## 2. Materials and Methods

### 2.1. Patient Selection

The probands for genetic testing were obtained from University Hospital Olomouc (Czech Republic), HEMACENTRUM Plzeň (Czech Republic), and University Hospital Martin (Slovak Republic) in 2016–2021.

The exclusion of exogenous factors and the fulfillment of at least one of the indication criteria were decisive for the indication for high-throughput genetic testing. We used the indication criteria described by Colucci and Tsakiris 2020 [15]—see Introduction.

We focused on indicated patients in whom the most common genetic risk factors, FV Leiden and G20210A, were first excluded. These patients had to repeatedly show low anticoagulant protein activity. Selected patients were enrolled in HTS, where the genes *SERPINC1*, *PROC*, *PROCR*, and *PROS1* (which encode key anticoagulant proteins) were scanned.

The activity of the proteins was examined using assays that were specific for each of them. Functional tests were performed at local clinics. Chromogenic assays primarily based on the addition of excess prothrombin or factor Xa were used to measure AT activity. The PC activity assay was performed using chromogenic substrate or activated partial thromboplastin time. The detection of PS activity was performed with a latex immunoassay as free PS (fPS). The cut-off values for functional assays are empirically determined and may vary slightly from institution to institution. At the University Hospital Olomouc, the cut-off values were as follows: AT < 75%, PC < 72%, and fPS < 53%.

In 2016–2021, HTS of candidate genes was performed in 37 patients. Not all of them correctly fulfilled the indication criteria and had to be excluded from the final results; therefore, these samples were used as HTS control data to check for possible sequencing artefacts. Three patients were family members of correctly indicated probands. The other 3 patients were primarily indicated for the detection of FV Leiden or G20210A.

Finally, after refining the indication criteria, 31 patients with anticoagulant protein deficiency were selected for the statistical evaluation of HTS.

### 2.2. DNA Isolation

The primary biological material for DNA testing was 9 mL of peripheral blood, which was collected in EDTA tubes. DNA was isolated from whole blood using the desalting method, which provides high quality DNA suitable for long-term storage. Alternatively, we performed isolation using the QIAcube automated isolator (www.qiagen.com accessed on 12 February 2022).

### 2.3. Ion Torrent High-Throughput Sequencing

We applied the amplicon sequencing methodology, which is suitable for the detection of point mutations, minor deletions, insertions, and duplications (approximately 20 bp), and if the targets are sufficiently covered, CNV analysis can also be applied.

The following manuals were used in particular sequencing steps: the Ion AmpliSeq Library Preparation on the Ion Chef System (MAN0013432) and the Ion 510, Ion 520, and Ion 530 Kit-Chef-Instructions for automated template preparation, chip loading, and sequencing (MAN0016854). All are available at (www.thermofisher.com accessed on 12 February 2022).

#### 2.3.1. Primer Design

Using the AmpliSeq Designer software (Ion Torrent™), amplicons were designed for the following genes: *SERPINC1*, *PROC*, *PROCR*, and *PROS1*. The range of amplicons was 125–275 bp. Ion AmpliSeq™ On-Demand panels were used for this study due to the reliability of target coverage. This approach allowed us to design amplicons with 100% coverage of coding sequences and exon/intron boundaries that potentially affected splicing.

#### 2.3.2. Library Preparation

Automated library preparation requires a low initial DNA concentration, usually 10 ng in a 15 µL volume. Libraries were prepared using the AmpliSeq™ Kit for Chef DL8 kit on the Ion Chef™ Instrument according to the instructions.

#### 2.3.3. Templating and Chip Loading

The Ion 510™/520™/530™ Kit-Chef was used for templating on the Ion Chef ™ Instrument. The Ion 520 chip was used for sequencing a smaller number of samples (8 samples), and the Ion 530 chip was used for more samples (16 samples). The sequencing run was scheduled on Torrent Suite for templating protocol at 400 bp.

#### 2.3.4. Sequencing

The Ion 510™/520™/530™ Kit-Chef was used for sequencing runs, and the Ion S5™ system was used for sequencing analysis. For sufficient end-to-end amplicon coverage, we set up 550 flows for each sequencing run. This HTS platform was provided by Thermo Fisher Scientific (www.thermofisher.com accessed on 12 February 2022).

#### 2.3.5. Data Analysis

Sequencing and coverage quality control were acquired from each sequencing run and sample. Amplicon coverage and overall sequencing data quality were assessed using Torrent Suite.

Sequencing instrument data were converted to the FASTQ format using the Torrent Suite software. At the next level, the aligned sequencing and variant data in the form of BAM or VCF files were uploaded to the Ion Reporter software for annotation. The Ion Reporter software allowed us to evaluate the clinical significance of the detected variants and filter these variants according to our requirements. The most important criteria were population frequency, PhyloP score, prediction software (PolyPhen, SIFT), the effect of variants on proteins, and previous classification in databases such as ClinVar. The classification of variants was conducted with respect to the ACMG/AMP criteria seen in the work of Richards et al. 2015 [17].

The evaluation procedure was based on checking for sufficient coverage (>20 reads per amplicon) and the visual inspection of variants using Integrative Genomics Viewer. Decisions regarding pathogenicity were based on the frequency of the detected minor allele (<0.01) and comparisons with the VarSome, ClinVar, and UCSC databases. In the case of a novel variant, we paid attention to the absence of alternative alleles according to gnomAD, 1000Gomes, prediction software (PolyPhen, SIFT, Mutation Taster, and PhyloP), VarSome database, and UCSC genomic browser.

### 2.4. MLPA Analysis

The on-demand panel design allowed us to perform CNV analysis. In this case, the MLPA method was used to confirm the detected CNVs. For MLPA analysis, we used the following probes: the SALSA MLPA P265 *PROC* probemix, the SALSA MLPA P112 *PROS1* probemix, and the SALSA MLPA P227 *SERPINC1* probemix.

### 2.5. Sanger Sequencing

Sanger sequencing was used to confirm the found causal variants and to predictively test family members at risk. For probands with a causal sequence variant detected by the Ion S5 platform, this was not necessary due to sufficient coverage and the mixed testing of patients with different diagnoses, which eliminated potential sample confusion.

### 2.6. Statistical Evaluation

To statistically evaluate the possible relationship between the values of functional tests in patients with and without a detected causal variant, a *t*-test (unpaired) was used. The null hypothesis was “no difference in functional test values between patients with and without the causal genetic variant”.

For the different number of functional tests recorded for each patient, the *t*-test was calculated from the median of the functional tests.

## 3. Results

Pathogenic or likely pathogenic variants were found in 21 of the 31 patients enrolled in the study. No causative SNV or CNV was found in 10 cases. The summed mutation detection rate (MDR) was 67.7%. No large deletion/insertion was detected.

We used the *t*-test to compare median functional test measurements in patients with and without causal genetic variants. A significant difference was found in patients with PS deficiency (*p* = 0.0047); see Table 1.

The results are summarized in Table 2 (patients with causal variant) and Table 3 (patients without causal variant). The overall data are summarized in Table 1. The distribution of functional test values in patients with or without the causal gene variant is shown in Figure 1.

In some measurements, higher-than-usual protein activity was repeatedly observed in congenital deficiencies. This was due to ongoing anticoagulation therapy, so these measurements were removed.

### 3.1. Antithrombin Deficiency

Eleven patients were indicated for the antithrombin deficiency phenotype, and causal variants were found in seven cases. The overall proportion of MDR was 63.6%. Most of the detected variants were missense (6) and nonsense (1). All the probable causal variants were found in the heterozygous state. One of these variants was novel—c.398A > T (p.Gln133Leu). We used the reference transcript NM_000488.4 to describe the variants in the *SERPINC1* gene.

### 3.2. Protein C Deficiency

The protein C deficiency indication was fulfilled in 13 patients, and the detection was successful in 10 cases. MDR was recorded in 76.9%. The found variants were missense (8), nonsense (1), and stop loss (1). Most of the described variants were in the heterozygous state (9), but we detected one homozygous variant. Three of the found variants were novel: c.715G > C (p.Gly239Arg), c.450C > A (p.Tyr150Ter), and c.866C > G (p.Pro289Arg). We used the NM_000312.4 reference transcript to describe the variants in the *PROC* gene.

### 3.3. Protein S Deficiency

For PS deficiency, 7 probands were indicated and 4 likely damaging variants were found. This represents an average MDR of 57.1%. We detected missense (2), frameshift (1), and splicing (1) causal variants, all in the heterozygous state. Two of the variants were newly discovered—c.1468delA (p.Ile490fs) and c.1931T > A (p.Ile644Asn). Visualization of the second variant see in Figure 2. We used the NM_000313.4 reference transcript to describe the variants in the *PROS1* gene.

## 4. Discussion

In our project, we focused on the HTS methodology, which is a valuable tool for the routine clinical diagnosis of undiscovered causal variants. Fidalgo and Ribeiro 2017 [18] reported the potential benefits of using HTS techniques to search for rare genetic variants in genes of interest. These methods should expand testing capacity, as the prevalence of patients with AT, PS, and PC deficiency in particular is likely to be underdiagnosed.

The HTS methodology has been successfully tested in this area of research. Lotta et al. 2012 [16] presented a pilot study that sequenced 186 selected genes and successfully identified previously known and novel potentially deleterious variants. Recent work has primarily focused on large panels of genes involving inherited bleeding disorders, thrombotic disorders, and platelet disorders. The most widely used methodology is capture-based exome sequencing (WES). In this project, we used amplicon-based sequencing because of its high coverage, low initial DNA input, and possibly better elimination of the *PROSP* pseudogene, which is highly homologous to the *PROS1* gene.

Researchers recently published the results of 300 patients sequenced according to a capture design for 63 genes and revealed 204 pathogenic or likely pathogenic variants (8 CNVs) [19]. A high MDR was observed in 64 patients—60.9% was reported in the WES for a thrombophilia panel extended by 55 genes [20]. Extensive results in this area were presented by Downes et al. 2019 [21], who provided HTS for 2396 patients suffering from bleeding, thrombotic, or platelet disorders (up to 96 genes) and revealed the molecular basis of the disorder in 37.3% of patients. A total of 284 patients were tested for thrombotic diagnosis, with potentially causative variants found in 48.9%. These papers were not exactly focused on MDR in cases of AT, PC, and PS deficiency. More specific results have mainly been provided in studies where molecular diagnostics was provided by Sanger sequencing [10,11,22,23,24,25,26,27,28,29,30,31,32,33,34,35].

This project was aimed at patients with the repeatedly detected deficiency of natural anticoagulant proteins. Of 31 referred patients, a causal variant was found in 21 cases, corresponding to an MDR of 67.7%. Similar data were published by Caspers et al. 2012 [23]—65.8%—and Kim et al. 2014 [25]—55.9%. Missense mutations were the most common, accounting for 76.2% of all detected causal variants. 

Using the *t*-test, we compared the values of functional tests in patients with and without the identified causal genetic variant. The aim was to assess the possibility of improving MDR, which could be influenced by shifting the cut-off activity of the monitored proteins. The results suggested a significant difference only in patients indicated for PS deficiency (*p* = 0.0047). Here, we could consider a possible threshold shift for patient indication for genetic testing. However, our cohort of patients with PS deficiency was small and requires further expansion. The correlation between MDR and AT, PC, and PS activity has already been addressed in the literature. Caspers et al. 2012 [23] reported high MDRs in patients and AT activity up to 75%. In the case of protein C activity, mutation detection was significantly reduced at levels above 60% (MDR of only 10%). In the case of PS, earlier studies reported that no mutations were found for activities above 53% and 55%, with the highest MDR values below 25% and 40%, respectively [10,23].

However, even in the case of PS deficiency, we do not consider shifting the cut-off for functional tests due to the main advantage of the used methodology because HTS offers an increase in assay throughput compared to the currently commonly used and time-consuming Sanger sequencing. Maintaining the current limits and using this HTS methodology may help to reduce the underdiagnosed cohort of patients with natural anticoagulant deficiency.

The presented approach enables automated library preparation, template creation, and sequencing using the System Ion S5™ platform. Patients with suspected hereditary anticoagulant deficiency can be combined with other diagnoses currently required by clinical genetics. This pipeline greatly simplifies the entire laboratory process.

### 4.1. Antithrombin Deficiency

Seven causative genetic variants were found in patients referred for AT deficiency. Our MDR result of 63.6% was slightly lower than the 74–83.5% reported in the literature [10,22,23,24,25]. Missense mutations were the most frequent at 85.7%; in previously published papers, this mutation was prevalent but slightly lower at 57.7–59% [10,26]. For the *SERPINC1* gene, we found that six out of seven variants were located in exon 2, and similar results have been previously described in the literature [10]. This is related to the involvement of the heparin-binding site, which is encoded by exons 2 and 3. Other authors have reported mutations in the gene in a rather linear fashion without mutational hotspots but with a slightly higher incidence at heparin-binding sites [24,27].

One novel missense variant c.398A > T (p.Gln133Leu) was found in this study. This variant is pathogenic because it corresponds to the phenotype of the patient. AT activity was repeatedly low (42–47%)—this corresponds to the position of the mutation on exon 2. The glutamine residue was highly conserved (PhyloP score = 7.6). This variant is not present in population databases and was classified as deleterious by the prediction software. Another variant of the same codon c.397C > A (p.Gln133Lys) was reported in a patient with a family history of thrombosis [28].

Interestingly, another found missense variant, c.79T > C (p.Trp27Arg), has rs1165816584 but no publication related to thrombotic condition. The patient with this variant had repeatedly low AT activity (60–63%). This variant is extremely rare and has only been listed in the TopMed population database. The variant is located in a conserved region and was evaluated as deleterious by the prediction software. A nonsense mutation c.80G > A (p.Trp27Ter) corresponding to AT deficiency was previously published at the same codon [27]. Other found variants have been previously published. Interestingly, the same missense variant c.236G > A (p.Arg79His) was found in two unrelated patients with a positive family history of thrombosis. The arginine residue is highly conserved and extremely rare. It is known as antithrombin Rouen with normal inhibitory activity but reduced heparin cofactor activity due to the substitution of arginine for histidine [29]. The causal effect of this variant has been confirmed by other studies that have found it in cohorts of selected patients [10,21]. Missense variants c.133C > T (p.Arg45Trp) and c.391C > T (p.Leu131Phe) have also been found in exon 2. Both have been described in the literature as causative [10,21,23,26]. We found one nonsense variant in exon 4 of the *SERPINC1* gene, c.685C > T (p.Arg229Ter). The proband had an extremely low functional assay level (24%), and the mutation was also found in other at-risk family members. This variant does not have an rs number, but Castaldo et al. 2012 [24] described it as the most severe phenotype in a cohort of 26 patients.

### 4.2. Protein C Deficiency

Thirteen probands were indicated for HTS scanning, and ten causal variants were identified, representing an MDR of 76.9%. In the case of a diagnosis of PC deficiency, previous reports have identified an MDR of 58.1–89.3% [10,22,23,25,30]. Of the 10 found variants, 8 (80%) were missense, 1 (10%) was nonsense, and 1 (10%) was stop-loss. One of the found variants was in the homozygous state. A similar condition—with a 76–84% predominance of missense mutations—has been described in previous studies [10,23,25,30]. The found variants were mainly in exon 9, where the serine protease domain is encoded. A slightly higher mutation rate was observed in this exon [10].

We identified three likely pathogenic/pathogenic novel variants in the *PROC* gene. The most interesting was a missense variant in exon 9—c.866C > G (p.Pro289Arg) in homozygous status. The proband (male) had repeated episodes of DVT and PE, but the first DVT was recorded at the age of 24 years. Functional test showed an activity of only 4.4–27.1%. The amino acid proline 289 is evolutionary conserved. We provided predictive testing for five other vulnerable family members. We found three homozygotes and two heterozygotes in this family. This variant is not present in population databases and was classified as deleterious by the prediction software. PC deficiency is usually transmitted but not always as an autosomal dominant trait. A variant at the same nucleotide position was previously published in a proband born to consanguineous c.866C > T (p.Pro289Leu) parents in a case of the incompletely recessive inheritance of the trait in the family, and a report of late-onset PC deficiency was previously published [31,32].

The second novel variant was a missense variant in exon 8—c.715G > C (p.Gly239Arg). The proband had a positive family history of thrombosis, and their protein activity was 49–57%. The mutation was positively detected in the brother. The glycine residue is highly conserved. This variant is not listed in the population databases and was rated as deleterious by the prediction software. The same variant at the protein level but a different nucleotide substitution c.715G > A (p.Gly239Arg) has been previously published, and this variant was also observed in the compound heterozygote state [21,25].

A third newly described variant was a nonsense in exon 6—c.450C > A (p.Tyr150Ter). The proband had a family history of thrombosis and was indicated for PC deficiency. The nonsense mutation resulted in a preliminary STOP codon in exon 6 of 9, presumably leading to protein damage.

Other found variants have been previously published. Missense mutations c.1301T > C (p.Val434Ala), c.1019C > T (p.Thr340Met), and c.1106C > T (p.Pro369Leu) were identified in exon 9. They have been previously identified in probands with DVT or a positive family history of thrombosis [10,30,33]. The stop-loss variant c.1384T > C (p.Ter462Gln) found in female probands has been previously described [23]. According to both VarSome and the nomenclature of the Human genome variation society the variant p.Ter462Glnext * 17 - the transition of cytosine for thymine led to a change in the STOP codon for the glutamine codon and ended at the new STOP codon at position 17. In exon 7, c.777A > T (p. Gln226Leu) was detected in two unrelated probands. Our last likely causal variant in the *PROC* gene was c.759C > A (p.His253Gln) in exon 8. Wypasek et al. 2017 [10] described these mutations in patients with DVT, PE, and a family history of thrombosis.

### 4.3. Protein S Deficiency

Seven probands with PS deficiency were examined, and four variants were found—the MDR value corresponded to 57.1%. Similar MDR values have been reported in previous studies, ranging from 37.8–52% [10,22,23,25]. When the cut-off change for fPS level was below 40%, the MDR increased to 77% [10]. Caspers et al. 2012 [23] reported a rapid decrease in MDR for PS activity >55%, and no mutation was detected in this cohort of patients. We found two missense, one frameshift and one splice variant, and this even distribution was probably due to the low number of patients. Missense mutations are the most frequent at 46–63% [10,11,23]. Kim et al. 2014 [25] reported no predominant mutation. In our small cohort of patients, we found no preferred region for mutation in the *PROS1* gene. A similar pattern was described in patients from the Polish Slavic cohort—the mutations detected were located across the entire gene [10].

We found a novel frameshift variant, c.1468delA (p.Ile490fs), in the *PROS1* gene with an observed fPS activity of 9–29%. This deletion in exon 12 (out of 15) resulted in a frameshift and, after the insertion of six amino acids, a premature STOP codon (p.Ile490LeufsTer6). The mutation was confirmed in the mother of a PS-deficient proband.

The second novel was the missense variant c.1931T > A (p.Ile644Asn). This transversion in exon 15 was found in a male proband with an episode of DVT at age 18. The proband had a positive family history of DVT on the maternal side of the pedigree. Laboratory results revealed recurrent low fPS activity (22.1–27.9%). This variant is not present in population databases and was classified as deleterious by the predictive software. Moreover, the isoleucine residue is evolutionarily conserved. A variant at the same position c.1931T > G (p.Ile644Ser) was previously described by Li and Long 1996 [34] in a family with PS deficiency and thrombotic disease. The found mutation occurs in the disulfide loop of PS. Li and Long 1996 [34] reported that this variant may result in a higher affinity between PS and the C4 binding protein, as well as lower levels of fPS.

Another found mutation was the splice variant c.1155 + 5G > A in intron 10. The Human Splicing Finder software predicted the effects of the variant, including the activation of a hidden donor site. The same variant was observed in a Korean patient with DVT [35]. In addition, this mutation was detected by HTS and is classified according to ClinVar as probably pathogenic [21]. In the same paper, a missense variant c.1916G > A (p.Cys639Tyr) was reported, and we found it in a proband with a low functional assay (18;23). García de Frutos et al. 2007 [11] reported this variant, noting that the potential loss of the Cys288–Cys568 bridge appears to be deleterious, and emphasized the importance of unique physiological properties, such as cysteine, for the laminin G-like domain.

## 5. Conclusions

This was a multicenter study from the Czech Republic and Slovakia focused on the application of HTS methodology for the detection of rare genetic variants in cases of the diagnosed deficiency of natural anticoagulant proteins. We observed MDRs similar to those reported in recent studies based on Sanger sequencing data.

In addition, we efficiently used the innovative and automated Ion Torrent sequencing platform, and this new approach allowed us to combine testing of patients with different diagnoses in a single sequencing run. This reduced processing time in the laboratory and maximized the speed of implementing a potential preventive regimen in the proband’s family.

The cut-offs of the functional tests by which patients were stratified for genetic testing were previously influenced by the difficulty of Sanger sequencing. Our data and the literature suggest that it is useful to expand the cohort of patients with significant clinical findings of thrombosis at a young age. Uncovering a causal genetic basis may have a beneficial effect on both the proband themselves and the prevention regimens for family members with a confirmed thrombophilic variant. In addition its application in anticoagulation therapy, a known causal genetic variant could also allow for preimplantation genetic diagnosis.

To eventually adjust the cut-off values of functional tests, it would be necessary to expand the group of tested patients. On the other hand, an excessive narrowing of the test criteria may lead to a reduced detection of patients with natural anticoagulant deficiency. The HTS methodology we have described is reliable, has a sufficient detection rate, and is suitable for the rapid and flexible testing of referred patients with natural anticoagulant protein deficiency. In the future, a larger cohort of patients will allow us to identify other possible high-risk genetic variants that may have been missed, e.g., due to a higher frequency in the population. The potential risk of variants should be assessed based on population studies and the family segregation of these variants with the observed phenotype. Risk variants might be expected, especially in a cohort with borderline functional test results.

## Figures and Tables

**Figure 1 diagnostics-12-01060-f001:**
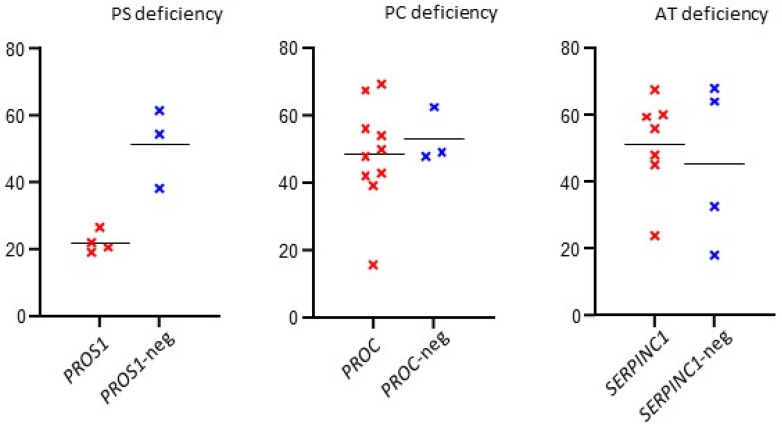
Distribution of functional test values in patients with or without a causal gene variants for specific indications—AT, PS, or PC deficiency. (*x*-axis: patients with a causal variant in a given gene—red marking/negative patients without a causal variant in a given gene—blue marking; *y*-axis: functional test value in %).

**Figure 2 diagnostics-12-01060-f002:**
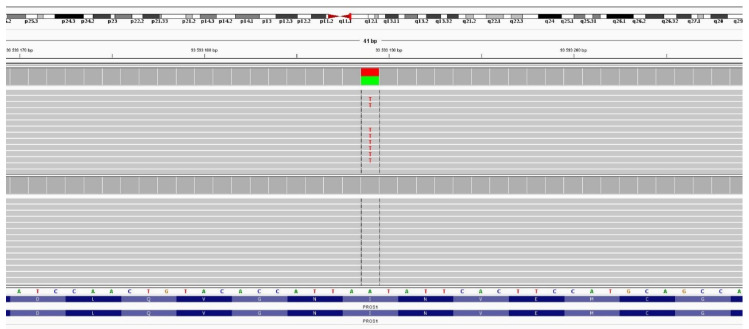
Visualization of the novel variant PROS1 c.1931T > A (p.Ile644Asn) using Integrative Genomics Viewer.

**Table 1 diagnostics-12-01060-t001:** A set of specifically indicated patients with an appropriate mutation detection rate.

Indication	Positive vs. Negative (*t*-Test)	Tested/(Variant Found)	Missense	Nonsense	Frameshift	Splicing	Stop Loss	Novel Variant
**ATD**	0.6241	11 (7) 63.6%	6	1	–	–	–	1
**PCD**	0.6286	13 (10) 76.9%	8	1	–	–	1	3
**PSD**	0.0047	7 (4) 57.1%	2	–	1	1	–	2
**Total**	**-**	**31 (21) 67.7%**	**16 (76.2%)**	**2 (9.5%)**	**1 (4.8%)**	**1 (4.8%)**	**1 (4.8%)**	**6 (28.6%)**

(ATD—antithrombin deficiency; PCD—protein C deficiency; PSD—protein S deficiency).

**Table 2 diagnostics-12-01060-t002:** Cohort of patients with a detected causal variant.

Proband Age (Sex)	Functional Test (%)	Functional Test Median (%)	Gene	Variant	Exon/Intron	Coordinate hg19	Variant Effect	Zygosity/Novel *	In Silico Predictions	Family History	Classification
1981 (F)	60; 61; 63	60	*SERPINC1*	c.79T > C p.Trp27Arg	2	1:173884020 rs1165816584	missense	het	damaging	na	likely pathogenic
1983 (F)	67; 68	67.5	*SERPINC1*	c.133C > T p.Arg45Trp	2	1:173883966 rs768704768	missense	het	damaging	no	likely pathogenic
1993 (F)	56	56	*SERPINC1*	c.391C > T p.Leu131Phe	2	1:173883708 rs121909567	missense	het	damaging	yes	pathogenic
1973 (M)	54; 65	59.5	*SERPINC1*	c.236G > A p.Arg79His	2	1:173883863 rs121909552	missense	het	damaging	yes	likely pathogenic
1971 (F)	48	48	*SERPINC1*	c.236G > A p.Arg79His	2	1:173883863 rs121909552	missense	het	damaging	yes	likely pathogenic
2004 (M)	42; 45; 47	45	*SERPINC1*	c.398A > T p.Gln133Leu	2	1:173883701	missense	het *	damaging	no	pathogenic
1932 (M)	24	24	*SERPINC1*	c.685C > T p.Arg229Ter	4	1:173879969	nonsense	het	_	yes	likely pathogenic
1980 (F)	50	50	*PROC*	c.450C > A p.Tyr150Ter	6	2:128180899	nonsense	het *	_	yes	likely pathogenic
1971 (M)	39; 45	42	*PROC*	c.677A > T p.Gln226Leu	7	2:128183802	missense	het	damaging	yes	pathogenic
1976 (F)	41; 45	43	*PROC*	c.677A > T p.Gln226Leu	7	2:128183802	missense	het	damaging	yes	pathogenic
1973 (F)	49; 56; 57	56	*PROC*	c.715G > C p.Gly239Arg	8	2:128184717	missense	het *	damaging	yes	pathogenic
1988 (F)	54	54	*PROC*	c.759C > A p.His253Gln	8	2:128184761 rs1458669732	missense	het	damaging	yes	likely pathogenic
1971 (F)	67; 68; 71; 71	69.5	*PROC*	c.1301T > C p.Val434Ala	9	2:128186437	missense	het	damaging	yes	pathogenic
1988 (F)	65; 70	67.5	*PROC*	c.1384T > C p.Ter462Gln	9	2:128186520 rs370298954	stop loss	het	_	yes	likely pathogenic
1978 (F)	47; 48; 59	48	*PROC*	c.1106C > T p.Pro369Leu	9	2:128186242 rs1211098698	missense	het	damaging/ tolerated	no	pathogenic
1953 (M)	4.4; 27.1	15.75	*PROC*	c.866C > G p.Pro289Arg	9	2:128186002	missense	hom *	damaging	yes	likely pathogenic
1994 (F)	39	39	*PROC*	c.1019C > T p.Thr340Met	9	2:128186155 rs766261022	missense	het	damaging	yes	pathogenic
1963 (M)	22	22	*PROS1*	c.1155 + 5G > A	10	3:93611772	splicing	het	_	yes	pathogenic
1981 (F)	9; 9; 11; 12; 19; 20; 23; 27; 29	19	*PROS1*	c.1468delA p.Ile490fs	12	3:93603596	Frameshift Deletion (INDEL)	het *	_	yes	likely pathogenic
1953 (F)	18; 23	20.5	*PROS1*	c.1916G > A p.Cys639Tyr	15	3:93593204	missense	het	damaging	yes	likely pathogenic
2002 (M)	22.1; 26.5; 27.9	26.5	*PROS1*	c.1931T > A p.Ile644Asn	15	3:93593189	missense	het *	possibly damaging/ damaging	yes	pathogenic

(Age—year of birth; M—male; F—female; het—heterozygote; hom—homozygote; *— novel variant; na—not available; in silico predictions—PolyPhen-2, SIFT; family history means history of thromboembolic states; classification with respect to ACMG/AMP—criteria for classifying variants).

**Table 3 diagnostics-12-01060-t003:** Cohort of patients without a causal variant.

Proband Age (Sex)	Functional Test (%)	Functional Test Median (%)	Gene	MLPA/CNV
1982 (F)	24; 30; 30; 31; 31; 32; 33; 35; 35; 36; 37; 38	32.5	*SERPINC1*	neg
1984 (F)	58; 60; 64; 66; 68; 68; 69; 70; 70	68	*SERPINC1*	neg
2019 (M)	18	18	*SERPINC1*	neg
1974 (F)	64	64	*SERPINC1*	neg
1999 (F)	44; 54	49	*PROC*	neg
2003 (M)	62; 63	62.5	*PROC*	neg
1993 (F)	41; 45; 48; 48; 49; 49	48	*PROC*	neg
1983 (F)	31; 45.6	38.3	*PROS1*	neg
1952 (M)	61.2; 61.9	61.55	*PROS1*	neg
1990 (F)	45; 64	54.5	*PROS1*	neg

(Age—year of birth; M—male; F—female; neg—negative).

## Data Availability

Given the focus, the published data are sufficient and the publication of additional data is not relevant.

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
