# Peer review of "Detection of Unknown and Rare Pathogenic Variants in Antithrombin, Protein C and Protein S Deficiency Using High-Throughput Targeted Sequencing"

_diagnostics, 2022, doi:10.3390/diagnostics12051060_

Round 1

Reviewer 1 Report

Authors report on the screening of a series of patients who suffered from venous thromboembolism with a suspect of a deficiency of one of natural anticoagulant.

Using a High-throughput platform they identified in 21 patients a pathogenic or likely pathogenic variant.

The study represents a valuable confirmation of previous reported approaches for the identification of variants in gene coding for natural anticoagulants.

In general, the study is well-conducted but it presents, in my opinion, some major flaws that need to be addressed.

  • the strategy used for the identification of the pathogenicity of gene variants observed is not reported. Did the Authors follow criteria of ACMG/AMP group?
  • was in silico prediction of the substitution for missense variants evaluated? ACMG/AMP criteria recommend the use. It would be worth indicating the prediction of some of these.
  • Authors should comment why they use a coagulative method to measure Protein S levels and not free antigen levels.

Minor points:

  • The definition Antithrombin III is no more used and the use of Antithrombin deficiency is recommended.
  • The meaning of the statistics reported in page 5 (lines 231-233) is quite obscure. Authors should rephrase it.
  • Frequencies indicated in the general population for the FV Leiden variant (2-15%) seems not realistic. More updated casistic should be quoted.

Author Response

Dear reviewer thank you for your valuable comments answering specific points : "Please see the attachment"

Reviewer 2 Report

The Authors evaluated a diagnostic approach based on a high throughput sequencing platform involving automated amplicon library preparation and target sequencing on the Ion Torrent platform. The analysis was performed on a group of 31 unrelated patients selected for
sequencing due to repeatedly low levels of one of the anticoagulant proteins. The topic is relevant and the manuscript is of interest, scientifically sound and with a robust methodology.

Major

  • Embolus does not generate only from deep veins of the lower extremities. And not only from the venous circulation. I would distinguish arterial and venous thromboembolism.
  • The presence of cancer increases the risk of death. Not sure is the presence of cancer, but rather the rate of cancer-related thromboembolism.
  • Authors state: ‘Environmental factors such as: surgery, use of oral contraceptives, immobilization, pregnancy, smoking, age, and cancer can trigger an episode of thrombosis’. Most of the factors mentioned above should not be defined as ‘enviromental’.
  • Line 52-53. I know it may be attractive to use the buzzer word COVID-19, but I would eliminate these lines.

Author Response

(The authors gave the same response as above.)

Round 2

Reviewer 1 Report

None.

Reviewer 2 Report

I do not have any further comment